# OpenReview forum: "Sharp Analysis for KL-Regularized Contextual Bandits and RLHF"
_NeurIPS.cc/2025/Conference — NeurIPS 2025 poster_

### Official Review · Reviewer_Rv2m · 2025-06-01

**Clarity:** 2
**Significance:** 3
**Originality:** 3
**Rating:** 4
**Confidence:** 4

**Summary:**

The paper studied contextual bandits and RLHF problem with a KL-regularized objective function, and the novelty is mainly on the theoretical side of sharp analysis for sample complexity. The paper designed a two-stage mixed sampling algorithm to achieve a good upper bound, which is the lower bound for sample complexity for contextual bandits and the analysis can be extended to RLHF to get good results.

**Questions:**

1. The assumption of boundness for reward is widely used in related literature, but I didn't see the necessity of the non-negative assumption for reward. Could you clarify it further?
2.  The authors discuss offline and online RLHF settings. But the writing is not clear, like which section or algorithm is for online and which part is for offline. For example, algorithm 1 seems like a hybrid setting. Could you discuss further?
3. In line 216, $\mathbb{E}_{x \sim d_0} \mathbb{E}_{a \sim \pi_o(\cdot \mid x)}\left[\left(R(\widehat{\theta}, x, a)-R\left(\theta_*, x, a\right)\right)^2\right]=O(1 / n)$ is typically called generalization error. I didn't get why the authors called them in-sample error. Could you give further explanation?

I really like the solid theoretical contribution of the paper. But I hope the authors have clearer writing to make the paper readable for more readers. Thus, depending on the rebuttal, I will consider changing my score.

**Ethical Concerns:**

["NO or VERY MINOR ethics concerns only"]

**Final Justification:**

The paper provided solid theoretical contribution. But the writing part should be improved and some concept should be cleaner to make the paper more readable. Overall, I'd like to maintain my score.

**Limitations:**

1. The author mentioned the reverse KL regularization in the abstract. It will be better if the authors discuss forward KL v.s.  reverse KL regularization somewhere.
2. I'd like to suggest that the authors give some proper citation for Definition 2.4.

**Quality:**

2

**Strengths And Weaknesses:**

Strengths:
The paper provided a solid theoretical contribution, and they also provided experimental results to verify their findings.

Weaknesses:
The writing and notation are confusing. For example, in line 217, what is $R(\theta_*,x,\pi^*)$? Please see the limitations part for more details.

---

> ### Author Response · Authors · 2025-07-31
>
> Thank you so much for your positive feedback! We answer your questions as follow.
>
> **Q1.** In line 217, what is $R(\theta_*, x, \pi^*)$?
>
> **A1.** Thank you for pointing it out. $R(\theta_\*, x, \pi^\*)$ here is the shorthand for $\mathbb{E}\_{a \sim \pi^\*(\cdot|x)}[R(\theta_*, x, a)]$. We define this short-hand notation in line 213. We will refer explicitly to where it is defined in the revision, to avoid any confusion for the reader.
>
> **Q2.** The assumption of boundness for reward is widely used in related literature, but I didn't see the necessity of the non-negative assumption for reward. Could you clarify it further?
>
> **A2.** Thank you for the question. You're right that the boundedness assumption on the reward is standard and plays an important role in ensuring generalization. As for the non-negativity assumption, it's not essential for the analysis itself, but we adopt it mainly for notational simplicity. We will remove the assumption in the revision.
>
> **Q3.** The authors discuss offline and online RLHF settings. But the writing is not clear, like which section or algorithm is for online and which part is for offline. For example, algorithm 1 seems like a hybrid setting. Could you discuss further?
>
> **A3.** Thank you for the helpful feedback. You're right that the distinction between the offline and online RLHF settings could be made clearer in the current version. Here's a clarification:
>   Thank you for the question. Algorithm 1 and Algorithm 3 are written in an abstract and general form to accommodate both online and hybrid scenarios. They do not explicitly specify how preference data is collected—this allows them to capture both offline and online instantiations:
>
> - In the hybrid setting, an offline dataset is fixed at the beginning and used throughout the algorithm.
>
> - In the online setting (which is the focus of our analysis), the dataset used in the first phase is actively collected via the policy $\pi_0$.
>
> To clarify further, our work does not directly study the offline RLHF setting. We mention it in the Introduction primarily to draw a contrast: our sample complexity bound depends only additively on the coverage coefficient, in contrast with prior offline results where the coverage plays a more limiting role.
>
> We will revise the manuscript to clarify these distinctions and better situate Algorithms 1 and 3 within the online setting.
>
> **Q4.** In line 216, $\mathbb{E}\_{x \sim d_0} \mathbb{E}\_{a \sim \pi_o(\cdot \mid x)}\left[\left(R(\widehat{\theta}, x, a)-R\left(\theta_\*, x, a\right)\right)^2\right]=O(1 / n)$ is typically called generalization error. I didn't get why the authors called them in-sample error. Could you give further explanation?
>
> **A4.** We use the term in-sample error to emphasize that this is the error measured on the same distribution as the training data, not over a distinct test distribution. That said, we agree that the term generalization error would also be appropriate here and may even be more standard. We will clarify this terminology in the revision to avoid confusion.
>
> **Q5.** The author mentioned the reverse KL regularization in the abstract. It will be better if the authors discuss forward KL v.s. reverse KL regularization somewhere.
>
> **A5.** Thank you for the suggestion. We agree that the distinction between forward KL and reverse KL regularization deserves discussion. In our work, we specifically adopt reverse KL regularization of the form $\text{KL}(\pi\|\pi_0)$ which encourages the learned policy $\pi$ to stay close to the prior policy $\pi_0$ but preserves modes and supports sharper optimization toward high-reward actions. This is particularly useful in preference-based learning, where identifying and focusing on optimal actions is crucial.
>
> In contrast, forward KL regularization, i.e., $\text{KL}(\pi_0(\cdot|x) | \pi(\cdot|x))$ penalizes deviation in the opposite direction and tends to promote coverage of the support of $\pi_0$. It is more conservative, often used when we want to prevent the learned policy from collapsing or ignoring actions present in the prior.
>
> We will add a brief discussion on this distinction in the main text to help the reader understand our choice of regularization and how it impacts the exploration and optimization dynamics in our setting.
>
> **Q6.** I'd like to suggest that the authors give some proper citation for Definition 2.4.
>
> **A6.** Thank you for the suggestion. We will revise the manuscript to include citations to relevant prior work[1].
>
> [1] Iterative Preference Learning from Human Feedback: Bridging Theory and Practice for RLHF under KL-Constraint.

---

> ### Comment · Reviewer_Rv2m · 2025-08-03
>
> Thanks for the detailed answers. The explanation of generalization error makes sense to me, and I agree that it should have a more appropriate name or add more explanation. And most of my concerns are solved. After reading the reviews from other reviewers, I would like to suggest that the authors improve the writing part, and I would like to keep my score.

---

> > ### Author Response · Authors · 2025-08-05
> >
> > We're glad that your questions have been resolved. Thank you for your support!

---

> > ### Comment · Area_Chair_KV52 · 2025-08-08
> >
> > Dear reviewer Rv2m,
> >
> > Thanks for contributing to the review process. May you also post a Mandatory Acknowledgement?
> >
> > Best, AC

---

> > > ### Comment · Reviewer_Rv2m · 2025-08-08
> > >
> > > Dear AC,
> > >
> > > Thanks for your time and this message. I didn't find the Mandatory Acknowledgement button in this paper. For other papers in my batch, there is a Mandatory Acknowledgement button. Could you help me check if there is any problem?
> > >
> > > Best,
> > >
> > > Reviewer Rv2m

---

> ### Comment · Reviewer_Rv2m · 2025-08-08
>
> Dear AC and authors,
>
> I think I've found the reason why I don't have a Mandatory Acknowledgement button....The authors used "official comments" not the "rebuttal" button to do the rebuttal.... Please tell me if there is a way to solve it.
>
>
> Best,
>
> Reviewer Rv2m

---

### Official Review · Reviewer_Cnt4 · 2025-07-01

**Clarity:** 3
**Significance:** 2
**Originality:** 3
**Rating:** 4
**Confidence:** 3

**Summary:**

This paper provides sharper sample complexity bounds for KL-regularized contextual bandits and RLHF. The authors propose a two-stage mixed sampling algorithm: in the first stage, the reward model and policy are updated using samples from a reference policy; in the second stage, the resulting intermediate policy is used to collect additional samples to estimate the optimal policy. With this approach, the algorithm achieves an O(1/e) sample complexity for small e, improving upon the standard O(1/e^2) complexity.

**Questions:**

1. How does the coverage coefficient $D^2$ scale with the size of the state-action space? Could the authors provide any additional comparisons (either analytical or empirical) with existing online methods in the literature regarding sample complexity?
2. Regarding Definition 3.5 (data coverage assumption for RLHF), the description is unclear. Could you provide additional explanation or intuition for the function $b(x)$? Is it dependent on $(\theta, \theta’)$?
3. While higher KL regularization coefficients lead to sharper sample complexity bounds in suboptimality gap, it is unclear whether this can be called the "power" of regularization. Since increasing regularization changes the optimal policy (closer to the reference policy), isn't this just a trade-off?
4. Minor issues:
- Line 258: In $$\pi: \mathcal{X} \to \mathcal{A}$$, should the policy be deterministic or can it be stochastic?
- Line 275: whether -> Whether
- Definition 3.1 (Preference Oracle): Is the output of the mapping P {0, 1} or [0, 1]? It is unclear whether P is the sampled output or the probability.

**Ethical Concerns:**

["NO or VERY MINOR ethics concerns only"]

**Final Justification:**

Based on the authors’ rebuttal and the strengths I mentioned in my initial review, I have decided to keep my score.

**Limitations:**

The paper does not sufficiently discuss its limitations, which should be more explicitly addressed.

**Paper Formatting Concerns:**

No formatting concerns identified

**Quality:**

3

**Strengths And Weaknesses:**

Strengths:
- The paper provides a thorough theoretical analysis, including tight upper and lower bounds, and covers both global and local coverage assumptions.
- The proposed two-stage mixed sampling algorithm is theoretically sound.
- The algorithm is easily implementable, requiring only one additional policy update and online sampling step.

Weakness:
- The provided sample complexity is not adequately discussed or compared with prior work. For example, the number of samples required in the initial stage may explode in practice due to its dependence on the coverage coefficient, potentially limiting the algorithm’s applicability only to asymptotical analysis (with extremely small epsilon).
- Empirical validation is limited, as the paper only compares the proposed method with an offline algorithm, where it is somewhat obvious that the online algorithm would perform better.
- The potential limitations of the proposed algorithm are not thoroughly discussed.

---

> ### Author Response · Authors · 2025-07-31
>
> Thank you for your positive feedback and insightful comments!
>
> **Q1.** How does the coverage coefficient $D^2$ scale with the size of the state-action space? Could the authors provide any additional comparisons (either analytical or empirical) with existing online methods in the literature regarding sample complexity?
>
> **A1.** The coverage coefficient $D^2$ characterizes the ability of the reward model to generalize from samples collected under $\pi_0$ to the ground-truth reward function. In general, $D^2$ tends to increase with the size of the state-action space, as a larger space typically requires more data for adequate coverage. However, $D^2$ is also influenced by the complexity and expressiveness of the reward function class, as well as its statistical generalization properties.
>
> As discussed in lines 211-221, existing theoretical analyses of online RLHF typically rely on optimism-based exploration and draw from classical bandit theory. In contrast, our analysis provides a sharper bound with **additive dependence** on the coverage coefficient $D^2$ and only **$1/\epsilon$ dependence** on the suboptimality gap. This yields a distinct theoretical perspective compared to prior work. For example, previous results in [1, 2] establish sample complexity bounds of $\tilde{O}(C_{\mathrm{gl}} \log|\mathcal{\Pi}| / \epsilon^2)$ and $\tilde{O}(d^2 / \epsilon^2)$, where $|\mathcal{\Pi}|$ corresponds to $|\mathcal{R}|$ in our setting—since each policy maps to a reward function up to a constant—and $d$ denotes the dimension of the reward function class.
>
> Thank you for the helpful suggestion. We will incorporate this discussion into the revision to clarify the role of the coverage coefficient $D^2$, its dependence on the state-action space and reward model class, and how our sample complexity bounds compare to existing online RLHF analyses.
>
> [1]  XIE, T., FOSTER, D. J., KRISHNAMURTHY, A., ROSSET, C., AWADALLAH, A. and RAKHLIN, A.
>  (2024). Exploratory preference optimization: Harnessing implicit q*-approximation for sample
> efficient rlhf. ICLR 2025.
>
> [2]  XIONG, W., DONG, H., YE, C., WANG, Z., ZHONG, H., JI, H., JIANG, N. and ZHANG, T.
>  (2024a). Iterative preference learning from human feedback: Bridging theory and practice for
>  rlhf under kl-constraint. In Forty-first International Conference on Machine Learning.
>
> **Q2.** Regarding Definition 3.5 (data coverage assumption for RLHF), the description is unclear. Could you provide additional explanation or intuition for the function $b(x)$
> ? Is it dependent on $\theta, \theta'$?
>
> **A2.**  Thank you for the question. Yes, the baseline function $b(x)$ can depend on the specific pair $(\theta, \theta')$. In the preference-based RLHF setting, we can only recover rewards up to a context-dependent constant shift, since preferences provide information about relative—not absolute—reward values. Consequently, we only need to learn the relative differences in reward values under a given reward model. The baseline $b(x)$ is introduced to account for this inherent identifiability issue, and thus, it is not necessarily independent of the specific pair $(\theta, \theta')$.
>
> We will clarify this point further in the revised version of the paper.
>
> **Q3.** While higher KL regularization coefficients lead to sharper sample complexity bounds in suboptimality gap, it is unclear whether this can be called the "power" of regularization. Since increasing regularization changes the optimal policy (closer to the reference policy), isn't this just a trade-off?
>
> **A3.** Thank you for raising this important point. You're absolutely right that increasing the KL regularization coefficient $\eta$ biases the optimal policy toward the reference policy $\pi_0$, and this represents a trade-off between exploration and exploitation. However, our use of the term "power" of regularization refers to the statistical efficiency it brings: a higher $\eta$ leads to tighter sample complexity bounds, particularly in terms of the suboptimality gap.
>
> This doesn't mean regularization is universally beneficial since it improves learnability at the potential cost of optimality. The key insight in our work is that KL regularization can reduce the complexity of the learning problem by smoothing the optimization landscape and controlling the variance of estimators, making it easier to learn with fewer samples. So while it is indeed a trade-off, our analysis quantifies the benefit side of that trade-off more precisely.
>
> We will clarify this interpretation in the revised manuscript to avoid any ambiguity.
>
> **Q4.** Line 258: In $\pi: \mathcal{X} \to \mathcal{A}$. Line 275: whether -> Whether. Definition 3.1 (Preference Oracle): Is the output of the mapping P {0, 1} or [0, 1]?
>
> **A5.** Thank you for pointing out these typos. We will carefully correct the identified typos in the revised version.

---

> > ### Comment · Reviewer_Cnt4 · 2025-08-07
> >
> > Thank you, the authors have addressed my concerns. I will keep the score.

---

### Official Review · Reviewer_Gj2b · 2025-07-01

**Clarity:** 3
**Significance:** 2
**Originality:** 3
**Rating:** 4
**Confidence:** 3

**Summary:**

This paper investigates the RLHF problem in the contextual bandit setting, with the goal of establishing tight sample complexity bounds. The authors analyze both the RL and RLHF settings, deriving matching upper and lower bounds for each. Notably, they demonstrate that incorporating a KL constraint into the problem yields a sample complexity that scales as $1/\epsilon$, rather than $1/\epsilon^2$ as in prior works.

**Questions:**

1. See weakness.
2. Could you give a more detailed comparison between the things in Appendix B and [1]?



[1] Zhao, H., Ye, C., Xiong, W., Gu, Q., & Zhang, T. (2025). Logarithmic Regret for Online KL-Regularized Reinforcement Learning. arXiv preprint arXiv:2502.07460.

**Ethical Concerns:**

["NO or VERY MINOR ethics concerns only"]

**Final Justification:**

Most of my concerns are addressed. Specifically, the analysis for partial converge is valuable so I raise my score to 4.

**Limitations:**

yes

**Paper Formatting Concerns:**

No formatting concerns

**Quality:**

3

**Strengths And Weaknesses:**

**Strengths**

The work proposes a tight sample complexity for RL/RLHF with KL-regularization. The paper is well written and all the theorems seem correct.

**Weaknesses**

1. Definition 2.5 is too strong. Essentially, by collect N data from $\pi_0$, we can immediately find $\theta$ such that $\mathbb{E}_{x, a\sim \pi_0}[(R(\theta, x, a) - R(\theta^\star, x, a))^2] \le c  \frac{\log |\mathcal{R}|/\delta }{N} $ with probability $1-\delta$. By taking $N$ as some log order term, the right hand side can be bounded by constant. In this regard, Definition 2.5 actually claim that by collecting log number of data, we can find $\theta$ such that $R(\theta, x, a) - R(\theta^\star, x, a)$ can be well bounded for all $x, a$. This is impossible in practice.
2. The results in the paper only holds when $\eta$ is small, as a small $\eta$ ensures that $\pi(a|x)/\pi^\star(a|x)$ can be well bounded given $R(\theta, x, a) - R(\theta^\star, x, a)$ is well bounded. This makes the proposed tight bound practically meaningless, considering that $\eta$ is usually a very small value – for example, it is typically set to 1e-3 or 1e-4 in PPO or GRPO.
3. The analysis under RL from preference feedback is actually trivial. Given that the reward is bounded by B, it is straightforward that using exp(B) times more preference feedback is equivalent to directly obtaining reward feedback.

---

> ### Author Rebuttal · Authors · 2025-07-31
>
> Thank you for your constructive feedback!
>
> **Q1.** **Definition 2.5 is too strong.** Essentially, by collecting $N$ data from $\pi_0$, we can immediately find $\theta$ such that
>   $$
>   \mathbb{E}_{x,a \sim \pi_0} \left[ \left(R(\theta, x, a) - R(\theta^*, x, a)\right)^2 \right] \leq c \cdot \frac{\log |\mathcal{R}| / \delta}{N}
>   $$
>   with probability $1 - \delta$. By taking $N$ as some log-order term, the right-hand side can be bounded by a constant.
>
>   In this regard, Definition 2.5 actually claims that by collecting a logarithmic number of data, we can find $\theta$ such that
>   $
>   R(\theta, x, a) - R(\theta^*, x, a)
>   $
>   can be well bounded for all $x, a$. This is impossible in practice.
>
> **A1.** This interpretation slightly misrepresents what Definition 2.5 claims. The definition does not imply that one can uniformly bound
> $R(\theta, x, a) - R(\theta^*, x, a)$
> for all $x, a$ using only a logarithmic number of samples from $\pi_0$. First, it is not proper to claim that $N$ is some log-order term. In practice, $\log|\mathcal{R}|$ is roughly in the same order as the number of parameters, which can still be quite large. Moreover, Definition 2.5 is quite standard in offline RL: the number of samples required to achieve small uniform error over all inputs typically scales with the complexity of the function class and the desired resolution. Although weaker assumptions are sometimes studied, Definition 2.5 is a common assumption for offline RL ([1, 2, 3]), especially for works aiming for tighter bounds. Also, $D^2$ in Definition 2.5 can be a large number when the action space is not well-covered, but our upper bound only has an additive dependence on $D^2$.
>
> We can also obtain results under **partial coverage**. By using pessimism ($\hat R(x,a) - R_*(x,a) \le 0,\forall x,a$) and denoting $\Delta=\hat R - R_*\le 0$ and $\pi_f(a|x)\propto \pi_0(a|x)\exp(\eta(\gamma\Delta(x,a) + R_*(x,a)))$, we can prove from (2.2) that by taking the derivative for $U(\gamma)=\sum_{a\in \mathcal{A}} \pi_f^\eta(a|x) (\Delta(x,a))^2$ on $\gamma$,
>
> $
> \frac{\partial U(\gamma)}{\partial \gamma} = \sum_{a\in\mathcal{A}} \gamma\cdot(\Delta(x,a))^2\pi\_f\^\eta(a|x) \cdot \eta\big(\Delta(x,a) - \mathbb{E}\_{\pi_f^\eta} \Delta(x,a)\big) = \gamma\Big[\mathbb{E}\_{\pi_f^\eta} [\Delta^3(x,a)] - \mathbb{E}\_{\pi_f^\eta} \Delta(x,a)\cdot \mathbb{E}_{\pi_f^\eta} [\Delta^2(x,a)]\Big] \le 0,$
>
> where the last inequality holds since $\Delta(x,a)\le0$. Therefore, we get
>
> $
>     Q(\pi\^\*) - Q(\hat\pi_{pessi}) \le \eta\mathbb{E}\_{x\sim d_0}U(\gamma) \le \eta\mathbb{E}\_{x\sim d_0}U(0) = \eta\mathbb{E}\_{x\sim d_0}\sum_{a\in \mathcal{A}} \pi\_\*(a|x) (\Delta(x,a))^2 \le \eta C_{partial}\mathbb{E}\_{x\sim d_0}\mathbb{E}\_{\pi_0}(\Delta(x,a))^2,
> $
>
> where $C_{partial}$ is the single-policy coverage. However, the partial coverage condition cannot enjoy the \textbf{additive} dependence in the sample complexity. We will add and discuss this result in the revision.
>
>
> **Q2.** The results in the paper only hold when $\eta$ is small. This makes the proposed tight bound practically meaningless, considering that $\eta$ is usually a very small value — for example, it is typically set to $10^{-3}$ or $10^{-4}$ in PPO or GRPO.
>
> **A2.** It's not accurate to claim that $\eta$ is usually as small as $10^{-3}$ or $10^{-4}$. A more realistic value is $\eta \in [0.01, 0.5]$, depending on the task and the algorithm.
>
> | Setting | Typical KL Coefficient ($\beta$) |
> |--------|-----------------------------|
> | OpenAI's PPO for LLMs| 0.02 - 0.1 |
> | Hugging Face's TRL | 0.1 |
> | KL annealing setups | Start with 0.01, increase over time |
> | GRPO (Deepseek Math) | 0.04 |
>
> InstructGPT (OpenAI, 2022)
> "Training language models to follow instructions with human feedback" Appendix C.4
> https://arxiv.org/abs/2203.02155
>
> TRL (Hugging Face's trl library)
> PPO implementation with KL regularization
> https://github.com/huggingface/trl
>
> DeepSeekMath: Pushing the Limits of Mathematical
> Reasoning in Open Language Models
> https://arxiv.org/pdf/2402.03300
>
> The KL regularization coefficient in these reports is often denoted as $\beta$ or $\gamma$. You might have looked for the wrong notation.
>
> **Q3.** Could you give a more detailed comparison between the things in Appendix B and [1]?
>
> **A3.** [1] considers the online learning setting, where the learner interacts with the environment sequentially and must balance exploration and exploitation. To achieve a logarithmic regret bound, they design an optimism-based exploration strategy that leverages KL regularization to control exploration adaptively over time. Their algorithm explicitly addresses the challenges of limited feedback and non-stationary information acquisition.
>
> In contrast, Appendix B of our work focuses on the active learning regime. Here, the learner is allowed to actively query reward labels without updating the model iteratively, and the goal is to minimize sample complexity while ensuring policy quality under KL regularization. The main difference lies in the type of interaction: online decision-making in [1] vs. selective querying in our setting.
>
>
> [1] Zhao, H., Ye, C., Xiong, W., Gu, Q., & Zhang, T. (2025). Logarithmic Regret for Online KL-Regularized Reinforcement Learning. arXiv preprint arXiv:2502.07460.
>
> [2] Ruosong Wang et al. What are the statistical limits of offline rl with linear function approxima-
> tion? In International Conference on Learning Representations (ICLR), 2021
>
> [3] Wei Xiong et al. Nearly minimax optimal offline reinforcement learning with linear function
> approximation: Single-agent mdp and markov game. In International Conference on Learning Rep-
> resentations (ICLR), 2023
>
> [4] Q Di et al. Pessimistic nonlinear least-squares value iteration for offline reinforcement learning. ICLR 2024.

---

> > ### Comment · Reviewer_Gj2b · 2025-08-05
> >
> > Thanks for the responses. Please make sure the new results with partial converge assumption are incorporated into the final manuscript.

---

> > > ### Author Response · Authors · 2025-08-06
> > >
> > > Thank you for taking the time to review our paper! We will certainly incorporate these additional results and discussions into the revision.

---

### Official Review · Reviewer_5V1N · 2025-07-03

**Clarity:** 3
**Significance:** 3
**Originality:** 3
**Rating:** 5
**Confidence:** 3

**Summary:**

The paper proves that for KL-regularized contextual bandits and RLHF, the correct sample complexity is of order $\tilde{\Theta}(1/\epsilon)$, not $\tilde{\Theta}(1/\epsilon^2)$ ignoring other factors. The lower bounds are proven via standard info-theoretic arguments, and the upper bounds are shown via proposing a two-stage approach. The first stage collects samples from the reference policy, and the second stage collects samples from the learned policy, where the learning happened from first stage's collected samples.

**Questions:**

1. In Rmk 3.9, the authors state that $e^B$ is probably unavoidable. Can the authors comment on whether the techniques from [1] can be adopted to remove such dependency?
2. What parts of this paper can be easily extended to general preference learning scenarios?
3. Related to weakness, can the authors comment on the (educated) conjectured "correct" dependency on the policy coverage?
4. What is the intuition behind the fast rate of $1/\epsilon$, in "layman's terms"? In learning theory, the intuition that I'm familiar with is regarding the curvature of the loss and mixability properties of the loss function. Do we have something similar here? If yes, then the authors should include such intuition in the main text.


[1] https://arxiv.org/abs/2502.00666

[2[ https://jmlr.org/papers/v16/vanerven15a.html

**Ethical Concerns:**

["NO or VERY MINOR ethics concerns only"]

**Final Justification:**

During the rebuttals, some additional concerns for the lower bounds came up. But given the correctness of other parts as well as the significance of the new upper bound is appealing. Thus for the time-being (2025.08.03), I maintain my score of 4, but leaning towards acceptance, and willing to increase pending further discussions if possible. (2025.08.08) After the rebuttal, the authors have clarified all the concerns that I had, and thus I raise my score to 5.

**Limitations:**

yes

**Quality:**

3

**Strengths And Weaknesses:**

**Strengths**
- Well-written

**Weaknesses**
- Lower bound w.r.t. policy coverage is lacking
- References to the Appendix are lacking in the main text, e.g., Appendix D (experiments).
- Moreover, discussions regarding the dependency of the coverage coefficient on $\epsilon$ are lacking. Is it always negligible to claim that it is ignorable w.r.t. $1 / \epsilon$? or can it be $\frac{1}{\epsilon} \log\frac{1}{\epsilon}$, and so the current bound is tight up to logarithmic factors?

---

> ### Author Rebuttal · Authors · 2025-07-31
>
> Thank you for your positive feedback and insightful comments! We answer your questions point-by-point.
>
> **Q1.** Lower bound w.r.t. policy coverage is lacking.
>
> **A1.** While our lower bound mainly focuses on the dependency on $\epsilon$, we can extend the hard case in Theorem 3.6 to show that additive dependency on coverage coefficient is necessary.
>
> WLOG, we assume that the coverage coefficient $C$ is an integer. In the new construction, we let the size of the action set $|\cal{A}| = C$. We still define the same reward function as $R(\theta, x, a)$ in the proof of Theorem 3.6, where the only difference is that $\theta$ is now a mapping from $\mathcal{X}$ to $\cal{A}$. Similarly, we can show by Pinsker's inequality that $\mathbb{E}\_{Unif(\theta)}\mathbb{P}\_\theta(\pi\_{\text{out}}(\theta(x_0))\le \frac{1}{|\mathcal{A}|}) \ge \frac{1}{2} - O(\sqrt{c^2T/M})$.
>
> Then we have $$
> \mathbb{E}\_{\theta \sim \text{Unif}(\Theta)} \mathbb{E}\_{\pi\_{\text{out}}}
> \left[
> \log \frac{\pi\_{\text{out}}(a \mid x)}{\pi^*(a \mid x)}
> \right]
> = \mathbb{E}\_{\theta \sim \text{Unif}(\Theta)}
> \frac{1}{M} \sum\_{x \in \mathcal{X}} \mathbb{E}\_{a \sim \pi\_{\text{out}}(\cdot \mid x)}
> \left[
> \\log \frac{\\pi\_{out}(a| x)}{\\pi\^\*(a|x)}
> \right]\\ \geq \mathbb{E}\_{\theta \sim \text{Unif}(\Theta)}
> \frac{1}{M} \sum\_{x \in \mathcal{X}} \mathbb{P}\_{\theta} \left( \pi_{\text{out}}(\theta(x)) \leq \frac{1}{|\mathcal{A}|} \right) \cdot
> \text{KL}(Unif(\mathcal{A})\|\pi\_\theta\^\*)
> \\ \geq \left( \frac{1}{2} - O(\sqrt{c^2T/M}) \right)
> O(\eta^2c^2).$$
>
> When $\epsilon$ is sufficiently small, we can set $c$ to $O(\sqrt{\epsilon / \eta})$ and show that $T = \Omega(\eta M /\epsilon) = \Omega(\eta \log |\mathcal{R}|/\epsilon \log C)$ where $C$ is the coverage coefficient, and $|\mathcal{R}|$ is the size of the reward function class. Also note that we need at least $\Omega(C \log |\mathcal{R}|/\log C)$ rounds to explore all the actions. Hence the final lower bound is $\Omega(\max(\eta \log |\mathcal{R}|/\epsilon \log C, C\log |\mathcal{R}|/\log C)) = \Omega(\eta \log |\mathcal{R}|/\epsilon \log C+  C\log |\mathcal{R}|/\log C)$, which nearly matches our upper bound.
>
> **Q2.** References to the Appendix are lacking in the main text, e.g., Appendix D (experiments).
>
> **A2.** Thank you for pointing this out. We will add necessary references in the revision.
>
> **Q3.** Moreover, discussions regarding the dependency of the coverage coefficient on $\epsilon$ are lacking. s it always negligible to claim that it is ignorable w.r.t. $1/\epsilon$
>  or can it be $\frac{1}{\epsilon} \log \frac{1}{\epsilon}$., and so the current bound is tight up to logarithmic factors?
>
> **A3.** In our setting, the coverage coefficient is not necessarily relevant to the value of $\epsilon$. Note that in our upper bound, we only have an additive dependency on the coverage coefficient (the value of $m$ in Theorem 3.8 only depends on $D^2$, $\eta$ and log-covering number of the reward function class). As explained in **A1**, we believe this dependency is not avoidable.
>
> **Q4.** In Rmk 3.9, the authors state that $e^B$ is probably unavoidable. Can the authors comment on whether the techniques from [1] can be adopted to remove such dependency?
>
> [1] Avoiding $\mathbf{exp(R_{max})}$ scaling in RLHF through Preference-based Exploration.
>
> **A4.** Thank you for mentioning this recent work. We notice that Assumption 4.3 in [1] requires that the underlying reward function is a linear function, which is strictly stronger than the assumption made in this paper. (We only requires that the reward function class has a finite covering number.) Since our paper firstly consider the sharper bound with respect to $\epsilon$, we believe that it would be a interesting future direction to consider remove $\exp(R_{max})$ dependence in our bound when the reward function class is linear.
>
> **Q5.** What parts of this paper can be easily extended to general preference learning scenarios?
>
> **A5.** The main sharper analysis can be easily extended to the general preference setting, since the key technique is Lemma 2.9. This lemma implies that for any functions $R^*(x,a)$, $R(x,a)$, the KL-regularized function satisfies
> $$
> V(\pi\^\*, R\^\*) - V(\pi\_R, R) \le \eta\, \mathbb{E}\_{\pi\_{R'}}\left( R(x,a) - R\^\*(x,a) \right)^2,
> $$
> where
> $$
> V(\pi, R) = \mathbb{E}\_{x \sim d\_0,\, a \sim \pi}\left[ R(x,a) - \eta^{-1} \mathrm{KL}(\pi \| \pi\_0) \right].
> $$
> This property follows from the strong convexity of the KL regularization, and thus also applies in the general preference setting, as the objective still includes a KL term.
>
> Moreover, according to [1], we can focus on the objective for the max player instead of analyzing the full Nash equilibrium. Therefore, it is promising that a similarly sharper bound can be obtained under general preference. Finally, if one considers uniform policy coverage (by extending Definition 2.5 to the general preference setting), this extension should be straightforward to accomplish.
>
> [1] Ye, C., et. al. Online iterative rein- forcement learning from human feedback with general preference model. NeurIPS.
>
> **Q6.** What is the intuition behind the fast rate of $1/\epsilon$, in "layman's terms"? In learning theory, the intuition that I'm familiar with is regarding the curvature of the loss and mixability properties of the loss function. Do we have something similar here? If yes, then the authors should include such intuition in the main text.
>
> **A6.** Thank you for your valuable insight. Yes, there is indeed a similar intuition behind the fast
> $1/\epsilon$ rate in this setting. Just as in learning theory where fast rates often arise from strong convexity or the mixability of the loss function, here the key driver is the curvature induced by the KL regularization. Specifically, the strong convexity of the KL divergence around the optimal policy ensures that the optimization landscape penalizes deviations more heavily, enabling tighter control over regret or value suboptimality.
>
> We will revise the paper to emphasize this intuition more in the main text.

---

> ### Comment · Reviewer_5V1N · 2025-08-04
>
> Thank you for your detailed response. I'm satisfied with most of the responses, but I have some follow-ups on the lower bounds. Apologies for the late response; I wanted to make sure that I'm not too wrong.
>
> **Continuing regarding A3.** After some consideration, I realized where my confusion was. In the lower bound statements (Theorem 2.6 & 3.6), you state them as there exists some instance of KL-regularized contextual bandit/RLHF with "$O(N\_{\mathcal{R}}(\epsilon))$ coverage coefficient." At least in the main text, this should be rewritten as a clearer statement to avoid any confusion.
>
> **Regarding the proof of Theorem 2.6 (and 3.6).**
> According the the construction and the proof, it states that $N\_{\mathcal{R}}(\epsilon) = M$ and $D^2 = O(M)$ (line 1060-1062). Honestly, I don't think both are trivial. For instance for the first part, isn't $N\_{\mathcal{R}}(\epsilon) = |\Theta| = 2^M$? Maybe I'm completely mistaken here, but to my understanding, the two reward functions, with $c = 64 \epsilon$ can't be within infty distance of $\epsilon$ unless $\theta = \theta'$? Also the second part, can you provide the computation that leads to $D^2 = O(M)$? Also what is the reference policy considered here? I think it is denoted as $\theta\_*$ and $\pi^*$ but never explicitly defined.
>
> **Continuing regarding A1** This is my understanding of your new proof. You use the simple fact that $|\mathcal{R}| = A^M$, and thus, $M = \frac\{\log|\mathcal{R}|\}\{\log |A|\}$. Then, the same proof as of Theorem 2.6 & 3.6 are used to obtain $T \gtrsim \frac{\eta M}{\epsilon} \wedge MA = \frac{\eta \log|\mathcal{R}|}{\log |A|} \wedge \frac{A \log|\mathcal{R}|}{\log |A|}$, where the last lower bound is trivial as you should explore all $MA$ context-action pairs (for the constructed instance? Correct me if I'm wrong). Then lastly, conveniently setting $|A| = C$ gives the coverage coefficient-dependent lower bound. Of course this isn't cheating from worst-case minimax perspective, but I feel that this is still bit cheating. Also, why is this not using the same argument as in the submitted manuscript that gives the second $M / \epsilon^2$?
>
> **Overall regarding the lower bound** Continuing on, I'm still a bit skeptical about the lower bound involving the coverage coefficient. This is because the instance that the authors have constructed is a very specific subclass of the problem instances with the coverage coefficient, obtained by explicitly controlling the action set size. I was expecting lower bound similar to [1, Theorem 2.1] or [2, Theorem 1.1], where the lower bound statement is something like "Fix context/response space size. For given (upper bound of) coverage coefficient, there exists an instance with the given coefficient such that the lower bound is ...".
>
> Of course in the context of structural bandits (e.g., linear bandits), the action space is chosen to be intentionally exponential in the dimension. But then, I believe that the nature of those lower bounds is to show that even though the spaces are exponential, *if there are additional structural assumption (linear)*, then one can avoid direct dependencies on the sizes of those spaces. In this submission, as the author has elaborated in my above response, they don't explicitly consider any additional structural assumption.
>
>
> [1] https://arxiv.org/abs/2503.07453
>
> [2] https://arxiv.org/abs/2111.10919
>
>
> **Finally...**
> As I may have missed some things and considering my late response, I will keep my score for the time being with some reservation, pending the authors' responses to other reviewers as well as subsequent reviewer discussions. Please feel free to let me know if there are anything that I missed or misinterpreted or mistaken in my response. Thanks!

---

> > ### Author Response · Authors · 2025-08-05
> >
> > Thank you for your further feedback. We address your questions as follow.
> >
> > **Regarding the ambiguity in Theorem 2.6.** Your understanding is correct. In Theorems 2.6 and 3.6, the coverage coefficient should be $O(M)$, which is $O(\log N_{\mathcal{R}}(\epsilon))$. There was a typo in the manuscript—we mistakenly omitted the $\log$ term in the $O$ notation. We will correct this in the revision.
> >
> > To show that $D^2 = O(M)$, recall the definition:$$D^2 := \sup_{\theta, \theta'} \sup_{x, a} \frac{[R(\theta, x, a) - R(\theta', x, a)]^2}{\mathbb{E}_{x' \sim d_0, a' \sim \pi_0(\cdot | x')} [(R(\theta', x', a') - R(\theta, x', a'))^2]}$$
> >
> > Taking $\theta, \theta'$ be a pair of reward function s.t. $\theta(x) = \theta'(x)$ for all $x \in \mathcal{X} \backslash \{x_0\}$ and $\theta(x_0) \neq \theta'(x_0)$, we have $D^2 \ge \frac{c^2}{\frac{1}{M} \cdot 1 / 2 \cdot c^2} = 2M$. On the other hand, observe that $D^2 \le \sup_{\theta, \theta'} \sup_{x, a} \frac{[R(\theta, x, a) - R(\theta', x, a)]^2}{\mathbb{P}(x_0)\cdot \frac{1}{2} (R(\theta, x, a) - R(\theta', x, a))} = 2M$. Therefore, $D^2 = 2M$.
> >
> > The reference policy $\pi_0$ is the uniform random policy. We will make it explicit in the revision.
> >
> > **Continuing regarding A1.** Thank you for the thoughtful summary and feedback. You are right that our new lower bound construction builds on the simple fact that the learner need to explore across the context-action space in our construction. Our goal was to isolate the dependence on the coverage coefficient $C$ as clearly as possible, and for that, we opted for the most interpretable hard instance that still reveals the minimax rate. You're right that setting $|\mathcal{A}| = C$ makes the bound match the upper bound in terms of the coverage coefficient. This might feel like “cheating,” but from the minimax perspective, it is valid: we are allowed to tune instance parameters to saturate the upper bound and show its tightness. In fact, we see value in such a simple instance-both as a sanity check and to convey the key intuition about why the coverage coefficient appears in the bound.
> >
> > **Overall regarding the lower bound.** To address your point about the linear bandit setting: you raise an excellent point. In linear bandits, it is harder to decouple the role of $C$ from the log covering number $\log N_{\mathcal{R}}(\epsilon)$, which typically scales with dimension $d$. As you suggest, one can construct a simple example where $\pi_0$ is uniform over standard basis vectors and the action set is the unit sphere, yielding $D^2 = \Theta(d)$. However, obtaining tight lower bounds in that setting is more subtle, since $\log N_{\mathcal{R}}(\epsilon)$ cannot be treated independently. We are currently exploring such directions for future work.
> >
> > We also note that in the construction of [1], it is more convenient to express the dependence on $C$ since the data can only be drawn from $\pi_0$, resembling an offline learning regime. As a result, their hard instance may involve fewer actions.
> >
> > Lastly, regarding the earlier $M / \epsilon^2$ term: in the sketch given in A1, we assumed $\epsilon$ is sufficiently small, so we omitted it for clarity. However, the bound still holds for larger $\epsilon$, and we can include this detail explicitly in the final version.
> >
> > We would be happy to address any further questions or clarifications you may have. Thank you again for your time and effort in reviewing our paper!

---

> > > ### Comment · Reviewer_5V1N · 2025-08-06
> > >
> > > Thank you for the further detailed answer. But I'm afraid that I'm still a bit confused here. Am I correct in saying that one can set the action space $A$ as large as one wants, way beyond $C$? Then it seems that one can construct arbitrarily high lower bound by arbitrarily increasing $A$. At first I thought that if you arbitrarily increase the action set, then the coverage coefficient of the constructed instance is not $C$, making it invalid. But then, the authors' argument shows that $D^2 = 2M$ regardless of the action set size. BTW by coverage coefficient, you mean $D^2$, not $C$, right?

---

> > > > ### Author Response · Authors · 2025-08-06
> > > >
> > > > Thank you for your follow-up questions.
> > > >
> > > > We would like to clarify the two different notations as follows. The constant $C$ refers to the global coverage coefficient defined in Definition 2.10. For the considered action set $\mathcal{A}$, $C$ is at least $|\mathcal{A}|$, so it is not invalid to increase the size of $\mathcal{A}$ arbitrarily.
> > > >
> > > > Regarding the quantity $D^2$ defined in Definition 2.5, we showed that $D^2 = 2M$ in the original construction used in the proof of Theorem 2.6, where $|\mathcal{A}| = 2$, which answers your original question. In the more general construction presented in the rebuttal, we have $D^2 = M |\mathcal{A}|$.

---

> > > > > ### Comment · Reviewer_5V1N · 2025-08-09
> > > > >
> > > > > I see thank you for clearing up the confusion! I have no further concerns or questions, and thus I raise my score. Please make sure to include/clarify the discussions in the camera-ready version.

---

### Decision · Program_Chairs · 2025-09-17

**Decision:**

Accept (poster)

**Comment:**

(a) Summary: This work provided sharper sample complexity bounds for KL-regularized contextual bandits and RLHF. The authors proposed a two-stage mixed sampling algorithm: in the first stage, the reward model and policy are updated using samples from a reference policy; in the second stage, the resulting intermediate policy is used to collect additional samples to estimate the optimal policy. With this approach, the algorithm achieves an $O(1/e)$ sample complexity for small e, improving upon the standard $O(1/e^2)$ complexity.

(b) Main reasons for decision/Cons: The work proposed a tight sample complexity for RL/RLHF with KL-regularization. The paper is generally well written.

(c) Change during rebuttal: The author(s) provided good response to allay the concerns from reviewers, especially the concerns about Definition 2.5 and Theorem 2.6 (lower bound). Some reviewers (e.g. Reviewer 5V1N) raised the rating during rebuttal. Such explanation should be included in the camera-to-ready version.

(d) Decision: I vote for accept with the average rating as 4.75.
- Most reviewers voted for 5 and are satisfied with the paper while only one reviewer (Reviewer Cnt4) feel that 'the paper does not sufficiently discuss its limitations, which should be more explicitly addressed.'